# The Effect of Primary Aldosteronism on Carotid Artery Texture in Ultrasound Images

**DOI:** 10.3390/diagnostics12123206

**Published:** 2022-12-17

**Authors:** Sumit Kaushik, Bohumil Majtan, Robert Holaj, Denis Baručić, Barbora Kološová, Jiří Widimský, Jan Kybic

**Affiliations:** 1Faculty of Electrical Engineering, Czech Technical University in Prague, Karlovo Náměstí 293/13, 12 000 Prague, Czech Republic; 2Penta Hospitals CZ, Nemocnice Ostrov, U Nemocnice 1161, 363 01 Ostrov, Czech Republic; 31st Faculty of Medicine, Charles University in Prague, Kateřinská 32, 128 21 Prague, Czech Republic; 4Centre for Hypertension, 3rd Department of Medicine, General University Hospital in Prague, U Nemocnice 504/1, 128 08 Prague, Czech Republic

**Keywords:** primary aldosteronism, essential hypertension, ultrasound images, wavelets, Haralick textures, XGBoost classifier

## Abstract

Primary aldosteronism (PA) is the most frequent cause of secondary hypertension. Early diagnoses of PA are essential to avoid the long-term negative effects of elevated aldosterone concentration on the cardiovascular and renal system. In this work, we study the texture of the carotid artery vessel wall from longitudinal ultrasound images in order to automatically distinguish between PA and essential hypertension (EH). The texture is characterized using 140 Haralick and 10 wavelet features evaluated in a region of interest in the vessel wall, followed by the XGBoost classifier. Carotid ultrasound studies were carried out on 33 patients aged 42–72 years with PA, 52 patients with EH, and 33 normotensive controls. For the most clinically relevant task of distinguishing PA and EH classes, we achieved a classification accuracy of 73% as assessed by a leave-one-out procedure. This result is promising even compared to the 57% prediction accuracy using clinical characteristics alone or 63% accuracy using a combination of clinical characteristics and intima-media thickness (IMT) parameters. If the accuracy is improved and the method incorporated into standard clinical procedures, this could eventually lead to an improvement in the early diagnosis of PA and consequently improve the clinical outcome for these patients in future.

## 1. Introduction

Arterial hypertension is one of the main treatable causes of cardiovascular diseases [1]. One explanation for this caution could be atherosclerosis acceleration [2]. The prevalence of hypertension in adults is high, reaching 30–35% [3]. In 90–95% of the so-called primary or essential hypertension (EH) cases the cause is unknown. In the remaining 5–10% of cases of so-called secondary hypertension, the cause can be identified. From an epidemiological perspective, primary aldosteronism (PA) is the most frequent form of secondary hypertension resulting from endocrine causes [4]. Its prevalence in a non-selected population of patients with arterial hypertension is about 6% [5,6,7], but the prevalence in the population of patients with moderate or severe arterial hypertension reaches 20% [8]. PA is defined by the excess production of the hormone aldosterone from the adrenal glands. Prolonged exposure to the high content of aldosterone has a deleterious effect on the cardiovascular and renal system and increases the risk of cardiovascular mortality [9]. The involvement of the vascular wall in PA is complex. Elevated aldosterone levels stimulate cell growth and induce hyperplasia or hypertrophy [10] in vascular smooth muscle cells, leading to a remodeling of the vessel wall and an increase in the tunica media-to-lumen ratio in small resistance arteries [11]. In addition, the overproduction of aldosterone triggers profound changes in the extracellular matrix, including collagen accumulation and changes in the structure of collagen fibers, leading to arterial stiffening and fibrosis [12]. Prolonged exposure to the high content of aldosterone has a deleterious effect on the cardiovascular and renal systems and increases the risk of cardiovascular mortality. PA can be treated by either surgical removal of the adrenal gland (adrenalectomy), or with drugs (aldosterone receptors blockers). It is, therefore, essential to diagnose the PA early before the target organ damage occurs.

Ultrasound examination enables the assessment of intima-media thickness (IMT) in the common carotid artery (CCA), which is a dependable predictor of atherosclerosis and its extent [13]. Multiple studies have confirmed a significant link between IMT and atherosclerosis risk factors, such as arterial hypertension, smoking, diabetes, dyslipidemia, obesity, age, and male gender [14,15,16]. A significant correlation was described between IMT and systolic blood pressure (BP) [17,18]. Compared to patients with essential hypertension, patients with primary aldosteronism more frequently exhibit increased arterial wall stiffness [19] and increased IMT of the CCA using ultrasound measurement [20].

Previous studies found a potential use of ultrasound texture for evaluating arterial injury and cardiovascular risk [21]. In this work, we evaluate the texture of the arterial wall in ultrasound images in order to automatically distinguish between patients with PA and those with EH, assuming that more advanced fibrosis of the carotid artery wall in PA would be visible in the ultrasound. For comparison, we also evaluate the significance of commonly available quantitative clinical characteristics and biochemical parameters (e.g., BP, heart rate, glucose, cholesterol levels, and intima-media thickness).

## 2. Materials and Methods

### 2.1. Study Population

A total of 118 subjects were enrolled in the study: including 33 patients with PA, aged 42–72 years; 16 with aldosterone-producing adenoma confirmed by surgery; 4 with idiopathic aldosteronism; 13 with unclassified form (refusal of further investigation or unsuccessful adrenal venous sampling); 52 patients with EH; and 33 control subjects. We excluded patients after carotid endarterectomy or percutaneous transluminal carotid angioplasty, patients with carotid plaques, and patients with poor quality of carotid arteries visualization by ultrasound. Hypertensive subjects were recruited from patients hospitalized at our department for diagnoses of resistant hypertension. Subjects were considered hypertensive if their clinic BP (mean of three different sphygmomanometric measurements, each performed on a different day after a washout period of at least 2 weeks if previously treated with antihypertensive drugs) was higher than 140/90 mm Hg.

Chronic antihypertensive therapy was discontinued at least two weeks before admission and patients were switched to the treatment with α-blocker and/or slow-release verapamil, if necessary. Patients previously treated with aldosterone antagonists were not included in the study. Normotensive controls were recruited from subjects without a history of hypertension or cardiovascular disease and free of antihypertensive medication, who were referred for carotid ultrasound because of other risk factors for atherosclerosis. Controls were considered normotensive if their systolic BP was lower than 140 mm Hg and their diastolic blood pressure was lower than 90 mm Hg. Current smokers were defined as those who smoked ≥1 cigarette per day or those who quit smoking less than 1 year before examination. Diabetes mellitus was defined as fasting blood glucose ≥7.0 mmol/L measured on two occasions. All subjects with dyslipidemia (total plasma cholesterol ≥ 6.0 mmol/L or high-density lipoprotein cholesterol ≤ 1.0 mmol/L or triglycerides ≥ 2.2 mmol/L) adhered to a diet and some of them were on lipid-lowering therapy. Each participant signed written informed consent and the study was approved by the local Ethics Committee of the General University Hospital in Prague on 26 June 2003 and 21 June 2012, respectively (approval number 62/12, 21 June 2012).

### 2.2. Blood Pressure Monitoring

Office BP was measured in the sitting position by using a standard sphygmomanometer before the participant underwent the ultrasound examination. In hypertensive patients, the latest BP values before discontinuation of their usual antihypertensive therapy were also recorded. Twenty-four hours of ambulatory BP monitoring was performed using an oscillometric device (SpaceLabs 90207 Medical, Richmond, WA, USA), which was set to measure BP every 20 min during the day (from 06:00 to 22:00 h) and every 30 min during the night (from 10:00 p.m. to 6:00 a.m.).

### 2.3. Laboratory

The screening for the diagnosis of PA was based on the elevated aldosterone-to-renin ratio (ARR) ≥50 ((ng/dL)/(ng/mL per h)) (plasma renin activity (PRA) and aldosterone levels measured after 2 h upright position), suppressed PRA (≤0.6 ng/mL per h) and elevated plasma aldosterone (≥15.0 ng/dL). The diagnosis of PA was confirmed by the absence of plasma aldosterone suppression after the salt-loading test (plasma aldosterone ≥ 8.0 ng/dL) [22]. Plasma aldosterone and renin activity were assessed by radio-immunoanalysis (Immunotech, Prague, Czech Republic). Blood biochemistry (sodium, potassium, urea, creatinine, total cholesterol, LDL cholesterol, HDL cholesterol, triglycerides, and glycemia) was analyzed using multi-analyzers (Hitachi 717, Boehringer Mannheim, Germany) in the institutional central laboratory. The diagnosis of EH was made after careful exclusion of main forms of secondary hypertension (PA, pheochromocytoma, Cushing’s syndrome, renal parenchymal disease, or renovascular hypertension).

### 2.4. Carotid Ultrasound and IMT Measurement

High-resolution B-mode carotid ultrasound was performed with a multi-frequency (5–10 MHz) linear-array transducer (Acuson Sequoia 512, Siemens Medical Solutions, Mountain View, CA, USA). The subject was in a supine position with the head tilted 45° in the direction opposite of the carotid being measured. Standardized longitudinal B-mode images of the far wall were obtained proximally from the tip of the flow divider (zero reference point). CB and CCA segments were defined between 0 and 10 mm and between 10 and 20 mm from this reference point, respectively (Figure 1a).

Using Meijer’s Carotid Arc, images were taken at two angles of 90° and 150° for the right carotid artery, and 210° and 270° for the left carotid artery (Figure 1b) [23]. The optimal image was frozen on the top of the R-wave of the QRS complex, saved in Dicom format, and written to a 640 MB magnetic optical disc.

The IMT measurements were performed offline. Frozen images were displayed on the screen of a computer using an automated edge detection software, Image Pro-Plus version 4.0 (Media Cybernetics, Silver Spring, MD, USA). On each image, the visualized blood-intima and media-adventitia boundaries of the far wall were marked with a computer-mouse-controlled caliper within the defined segment. The largest distance between these two lines was considered representative IMT for each segment [13]. The average of 4 IMT measurements, the mean of maximum (mean-max) IMT (2 angles, 2 sides), was calculated for CCA and CB segments (CCA-IMT and CB-IMT). Combined IMT per patient was calculated as the average of CCA-IMT and CB-IMT.

Ultrasound examinations and the off-line IMT measurements were performed by a single sonographer and reader in one person (R.H.) blinded to participants’ diagnoses. Ultrasound examinations were performed in duplicate in 20 subjects within 3 weeks. IMT reproducibility quantified by interclass correlation coefficient was 0.86 for the distal segment of CCA and 0.79 for the segment of CB.

### 2.5. Texture Analysis

The block diagram of the proposed method is given in Figure 2 with the automatic pipeline shown with green background (Note that we are using one classifier based on texture features. The other pipelines shown in the diagram were implemented as a baseline for comparison).

During preprocessing, the images were cropped and intensity normalized and the white dot markers were masked and inpainted. We analyzed the far wall of the common carotid artery in which IMT measurements were performed. That is the segment extending from 10 to 20 mm from tip of the flow divider. The IMT measurement procedure is described in Section 2.4 (Carotid ultrasound and IMT measurement). Patients with atherosclerotic plaques were excluded from the study. The polygon representing the area of interest (ROI) was positioned at the location of the thickest intima-media complex so that blood-intima and media-adventitia boundaries of the far wall would be clearly visible. The minimum polygon length was 10 mm (see Figure 3).

The ROI was drawn by a technician (S.K.) using a custom software. The ROI copied the boundaries selected by the program Image Pro-Plus and was afterwards checked by a physician experienced in assessment of ultrasound findings (B.M.) to minimize the observer variability. Inside the region of interest, we calculated 152 texture features summarized in Table 1 and described in more detail below.

The first and simplest two texture features are the intensity mean and standard deviation.
$$\mu =\frac{1}{\left|Ω\right|}\overset{}{\underset{i\in Ω}{\sum }}f\left(i\right){\;}^{}$$
$$\sigma ={\left(\frac{1}{\left|Ω\right|}\overset{}{\underset{i\in Ω}{\sum }}{\left(f\left(i\right)-µ\;\right)}^{2}\right)}^{1/2}$$
where *f*(**i**) is the image intensity in pixel **i**, Ω is the set of pixels of the ROI and |·| denotes the number of elements.

Second, there are 140 Haralick texture features *H_d_*_,*θ*,*a*_, probably the most frequently used texture descriptors ever, which were shown to work well in a variety of applications [24]. To calculate *H_d_*_,*θ*,*a*_, we first quantize the image intensities *f* into *N_g_* uniform bins (we use *N_g_* = 16). The gray-level co-occurrence matrix (GLCM)
$$P_{d,\theta \;\;}\left(a,b\right)=\frac{1}{Z}\;\left|\left\{i,j\in Ω;\;f\left(i\right)=a,\;g\left(j\right)=b,\;\left|i-j\right|1=d,\;∠\left(i-j\right)=\theta \right\}\right|$$
counts the number of times that quantized pixel intensities *a* and *b* occur in two-pixel locations **i**, **j** within the ROI, such as the distance between these pixels is *d* ∈ {1, 2,…, and 5} in the direction *θ* ∈ {0°, 45°, 90°, and 135°} (corresponding to up, diagonally up right, right, etc.)

Each *P_d_*_,*θ*_ matrix is normalized by *Z* such that ∑*_a_*_,*b*_
*P_d_*_,*θ*_ (*a*, *b*) = 1 and then the following 7 scalar measures are evaluated using different aggregation operators *a* ∈ {1, 2,…, and 7}:•Inverse difference moment
$$H_{d,\theta ,1}=\overset{}{\underset{a,b;a\neq b}{\sum }}\frac{P_{d,\theta }\left(a,b\right)}{{\left|a-b\right|}^{2}}$$
•Correlation
$$H_{d,\theta ,2}=\overset{}{\underset{a,b}{\sum }}\frac{\left(a-\mu_{a}\right)\left(b-\mu_{b}\right)P_{d,\theta }\left(a,b\right)}{\sigma_{a}\sigma_{b}}$$
with
$$\mu_{a}=\overset{}{\underset{a,b}{\sum }}aP_{d,\theta }\left(a,b\right),\;\mu_{b}=\overset{}{\underset{a,b}{\sum }}bP_{d,\theta }\left(a,b\right)$$
$$\sigma_{a}^{2}=\overset{}{\underset{a,b}{\sum }}{\left(a-\mu_{a}\right)}^{2}P_{d,\theta }\left(a,b\right)\;,\;\sigma_{b}^{2}=\overset{}{\underset{a,b}{\sum }}{\left(b-\mu_{b}\right)}^{2}P_{d,\theta }\left(a,b\right)$$

•Contrast


$$H_{d,\theta ,3}=\overset{}{\underset{a,b}{\sum }}{\left|a-b\right|}^{2}P_{d,\theta }\left(a,b\right)$$


•Maximum


$$H_{d,\theta ,4}=\underset{a,b}{\max}P_{d,\theta }\left(a,b\right)$$


•Energy


$$H_{d,\theta ,5}=\overset{}{\underset{a,b}{\sum }}P_{d,\theta }{\left(a,b\right)}^{2}$$


•Dissimilarity


$$H_{d,\theta ,6}=\overset{}{\underset{a,b}{\sum }}\left|a-b\right|P_{d,\theta }(a-b)$$


•Entropy


$$H_{d,\theta ,7}=\overset{}{\underset{a,b}{\sum }}P_{d,\theta }(a,b)\log P_{d,\theta }\left(a,b\right)$$


The final 10 texture features are based on Haar wavelet frames [25]. The descriptors can be calculated rapidly using a filter-bank and also perform well for a number of applications. The Haar wavelet is defined by a low-pass filter *H*(*z*) = (1 + *z*^l^)/2 and a high-pass filter *G*(*z*) = (*z*^l^ − 1)/2, where *z* is the z-transform argument and *l* = 2^i^ is the filter size. In the time domain, the filters will be denoted [*h*]_↑*2*_*^i^* and [*g*]_↑*2*_*^i^*, where ↑ denotes upsampling. Starting from the input image *f*^(0)^ = *f* at resolution *i* = 0, we recursively obtain one low-pass and three high-pass sub-bands for *i* ∈ {1, 2, and 3}:$$f^{\left(i+1\right)}={\left[h_{x}\right]}_{↑2^{i}}*\;{\left[h_{y}\right]}_{↑2^{i}}*f^{\left(i\right)}$$
$$d_{1}{}^{\left(i+1\right)}={\left[g_{x}\right]}_{↑2^{i}}*\;{\left[h_{y}\right]}_{↑2^{i}}*f^{\left(i\right)}$$
$$d_{2}{}^{\left(i+1\right)}={\left[h_{x}\right]}_{↑2^{i}}*\;{\left[g_{y}\right]}_{↑2^{i}}*f^{\left(i\right)}$$
$$d_{3}{}^{\left(i+1\right)}={\left[g_{x}\right]}_{↑2^{i}}*\;{\left[g_{y}\right]}_{↑2^{i}}*f^{\left(i\right)}$$
where subscripts *x*, *y* indicate along which direction the filter was applied. Note that thanks to separability, only six 1D filtering operations are required, with kernels consisting of only two non-zero elements. The filtering is therefore extremely efficient. The wavelet texture features *W* are then defined as simple energy means.
$$W_{i,j\;}=\frac{1}{\left|Ω\right|}\;\|d_{j}^{\left(i\right)}\|{}^{2}$$
$$W_{0\;}=\frac{1}{\left|Ω\right|}\;\|f_{}^{\left(3\right)}\|{}^{2}$$
where *W*_0_ corresponds to the mean low-pass energy and the *W_l_*_,*j*_ to the mean high-pass energy at band *i* and orientation *j* ∈ {1, 2, 3}.

### 2.6. Statistical Analysis

Data were analyzed using Statistica software vers. 12.1 (StatSoft, Tulsa, OK, USA) and by custom software written in the Julia programming language (see www.julialang.org, accessed on 6 May 2022). Using both clinical parameters and features derived from the ultrasound images (see Table 1 for an overview), differences (i) between the PA and EH groups and (ii) between the PA + EH and control groups were investigated.

The statistical significance of the features (quantitative biomarkers) is evaluated using Welch’s unequal variance two-sample *t*-test, two-sample Kolmogorov–Smirnov test, and the Mann–Whitney–Wilcoxon rank-sum (*U*) test. The results are listed in Section 3.3. To quantify the relationship between the features and the outcome, selected subsets of the features are used to train an XGBoost classifier [26] for both tasks (i) and (ii) between the PA + EH and control groups (XGBoost is an efficient implementation of gradient boosting, which builds the classifier incrementally and stochastically as a linear combination of decision trees. It has been successfully applied to a large number of machine learning tasks). The classification accuracy is evaluated using the leave-one-out approach. Given that several images per patient are collected, the predictions are aggregated for each patient by averaging, i.e., taking the majority vote. At training time, the training samples are weighted so that each patient has a unit weight. In each leave-one-out iteration, images from the patient being tested are not used for training.

## 3. Results

### 3.1. Clinical Characteristics of Study Groups

The basic characteristics of the studied groups are shown in Table 2. The patients with PA, patients with EH, and normotensive subjects were matched for demographic characteristics, while both hypertensive groups showed similar BP values on their chronic antihypertensive medication. Patients with PA had borderline higher weights than controls. Controls had significantly higher high-density lipoprotein cholesterol than patients with PA and EH and slightly higher total cholesterol than patients with PA. Both hypertensive patient groups had significantly higher IMT measured in the distal segment of the CCA than control subjects. There was also significantly higher IMT of the CCA in patients with PA compared to patients with EH. The difference remained statistically significant even after the adjustment for age and mean 24 h systolic BP (*p* = 0.001). On the contrary, IMT measured in the CB segment was not significantly different between both hypertensive groups.

Table 3 compares antihypertensive treatment between PA and EH patients. There was no significant difference in the amount and type of antihypertensives used before and after discontinuation of the long-term treatment. The values of systolic and diastolic BP and total hypertensive burden as expressed by 24 h BP monitoring remained comparable between the two groups after switching to an α-blocker and/or slow-release verapamil. In addition, the estimated duration of hypertension was comparable in both groups.

### 3.2. Statistical Significance Testing

Table 4 shows clinical and IMT-related features (biomarkers), including aldosterone and renin measurements, used in order to distinguish between PA and EH, BP, and IMT, expectedly with significant differences between the EH and PA groups or PA + EH and control groups at *p* < 0.05. Furthermore, hypertensive and normotensive groups differed in BMI and cholesterol levels, especially HDL cholesterol levels. These findings are consistent with previously published papers, including our own works [20,27].

Most of the texture features (111 out of 150) had a significantly different mean between the PA + EH and control groups according to all three statistical tests (Table 5). However, none of the texture features alone could distinguish between essential hypertension and primary aldosteronism groups (at the 5% significance level).

### 3.3. Classification Performance

Table 6 shows the per-patient leave-one-out classification accuracy for different feature subsets for the two binary classification tasks of distinguishing between (i) PA versus EH and (ii) PA + EH versus controls. The clinical characteristics consisted of clinical and biochemical values (age, sex, systolic and diastolic BP, plasma glucose level, plasma cholesterol, HDL and LDL cholesterol level, smoking, and level of triglycerides). As expected, it is well possible to distinguish hypertensive subjects from controls using clinical features. The texture feature performance was weak (see the ROC curves in Figure 4).

On the other hand, note that texture features alone allow us to distinguish PA versus EH with an accuracy of 73%, which is better than 57% prediction accuracy using clinical characteristics alone or 63% accuracy using a combination of clinical characteristics and IMT parameters. The aldosterone-related biochemical measurements (plasma aldosterone, plasma renin activity, and ARR, referred to as “Aldo” parameters) serving as the gold standard for the diagnosis of PA achieve 92% accuracy (Table 6).

## 4. Discussion

Diagnosis based on ultrasound texture has been used in the case of various parenchyma organs, e.g., the thyroid gland [28] or the liver [29]. However, diagnosis using the ultrasound image structure of the vessel has been described rather sporadically in previous papers. These works focused only on either gray level distribution and the spatial relationships of intima-media complexes [30] or the texture features of the intima and media layers in patients with EH [31]. We showed that using the combined texture features only, we can distinguish between PA and EH subjects with an accuracy of 73%, which is better than 57% prediction accuracy using clinical characteristics alone or 63% accuracy using a combination of clinical characteristics and IMT parameters used in the current clinical process of early detection of PA.

The accuracy of endocrinological parameters which are used as a gold standard for PA diagnosis reached 92%. The explanation for why the accuracy did not reach 100% is that the diagnosis of PA is complex. The diagnosis does not consist of a single cut point but a range of values and not all patients meet all three diagnostic criteria (i.e., high aldosterone levels, suppressed renin activity, and an elevated ARR). In some patients with PA, especially at the beginning of the disease, aldosterone levels may not exceed the upper limit of the normal range value or some of these values may be in the gray zone [32]. In these patients, additional ancillary tests such as aldosterone and renin values in a supine position and suppression tests need to be performed to definitively confirm the diagnosis of PA.

The differences in individual texture descriptors between the two groups of hypertensive patients were not statistically significant. One of the possible explanations is that the fibrotization of the vascular wall is primarily caused by the overall BP-load. Since the pressure load is the same in both groups of hypertensives it can be assumed that the vascular changes caused by high pressure alone will be the same in both groups of patients and the additive effect of elevated aldosterone levels forms only a minority of the changes.

Ultrasound images have some built-in noise which may have a negative impact on classification results. Nevertheless, we believe that noise also contains an important part of the information we are trying to evaluate. Therefore, in our work, we evaluated the complex signal including the noise, because we suppose that by suppressing it, we might lose an important part of the information, and we think that the only way to judge the performance is based on the final classification results of the whole system.

We must consider the diagnosis of PA in patients with clinical and humoral characteristics (typically resistant hypertension and hypokalemia). The diagnosis is based on endocrinological parameters. Nevertheless, the estimation of these parameters, e.g., the renin activity, is difficult to determine and is not routinely assessed by every biochemical laboratory and definitive diagnoses should be confirmed in specialized centers for hypertension. Therefore, there may be a delay in the diagnosis of this disease. According to our study, diagnoses of PA are determined after 11.5 years of treatment for resistant hypertension on average. In other words, the patients with PA are insufficiently treated and suffer from high blood pressure for 11.5 years. Which is why we try to shift the focus of clinicians who treat patients with resistant hypertension towards certain commonly available examinations which might show pathological findings caused by resistant hypertension and therefore lead to earlier diagnoses of PA as the most common form of resistant hypertension [8]. In our work we used both clinical features (clinical characteristics and humoral values) as well as image texture features. This is because, unlike the renin estimation, an ultrasound examination of the carotid arteries is a commonly available examination.

Our study included patients with EH who had been referred to our center for exclusion of suspected secondary hypertension. The relatively lower age in the group with EH when compared to the normal hypertensive population could be due to the selection for referral to our center. Older hypertensive patients may not be referred to our center as often, for example, due to a lower probability of secondary hypertension diagnosis.

We are aware of some limitations of our work.

First of all, it is a retrospective proof-of-concept study conducted on data used for other purposes. The ultrasound images were originally intended to be used to measure the distance between two optical interfaces by automatic software and, therefore, were not focused on capturing all the features in the best possible quality; hence, a prospective study with the same setting of all ultrasound image parameters will certainly be necessary to validate our study.

Second, the classification accuracy still needs to be improved in order to be clinically useful, which should be possible by training the classifier and feature extraction on a bigger dataset. Apart from the size of the dataset, other factors contributing to lower classification accuracy are the small thickness of the ROI (sometimes 6–8 pixels only) and the annotation of the ROI. The inconsistency in manual annotation can be resolved by the automatic localization method, which is one of the fundamental points of image processing, and optimizing texture analysis and classification. This process has already been described in previous works [33].

Third, this method allows us to segregate IMTs from background and surrounding tissues and to identify and exclude atherosclerotic plaque from IMT evaluation. Furthermore, it could pick multiple ROIs per image, thereby evading the small thickness problem too.

We expect that these changes should lead to further improvement in classification accuracy. The automatic texture evaluation could then be performed for all carotid artery ultrasound examinations, serving as an essentially ‘free’ screening procedure for PA, increasing the chances of its detection and, thus, leading to an earlier diagnosis and an appropriate treatment.

## 5. Conclusions

We have described an automatic method for quantitative evaluations of the vessel wall texture in longitudinal ultrasound images of the carotid artery. The changes in the texture of the carotid wall in hypertensive patients could help to suggest a diagnosis of PA during routine carotid ultrasound examinations. Although the prediction accuracy is currently too low to be useful in clinical practice, we have shown that it is possible to approximately distinguish differences in the texture of the carotid wall between EH and PA.

If incorporated into standard clinical procedures, this could eventually lead to an improvement in the early diagnosis of PA and consequently improve the long-term clinical outcome for these patients.

## Figures and Tables

**Figure 1 diagnostics-12-03206-f001:**
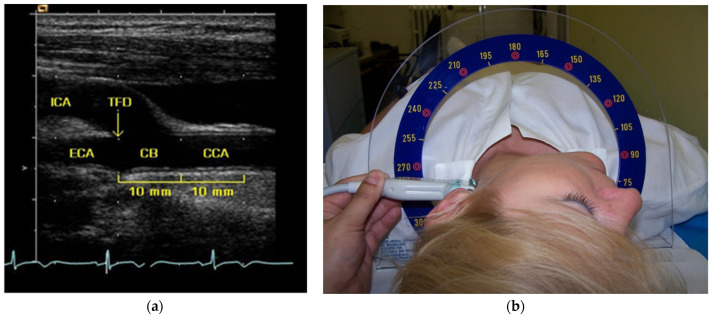
(**a**) Longitudinal B-mode image of the carotid artery showing clear interfaces for measurement of intima-media thickness at the far wall of the common carotid artery and carotid bifurcation. CB, carotid bifurcation; CCA, common carotid artery; ECA, external carotid artery; ICA, internal carotid artery; TFD, tip of the flow divider. (**b**) The Meijer´s Arc allows a standardized scan of the left and right carotid artery in predefined angles.

**Figure 2 diagnostics-12-03206-f002:**
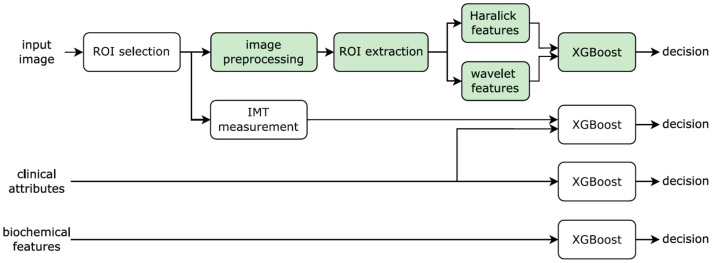
Block diagram of the proposed methodology. The new parts that our method consists of are framed in green, while the steps done previously or done only for the experimental comparison have a think frame. ROI, region of interest; IMT, intima-media thickness.

**Figure 3 diagnostics-12-03206-f003:**
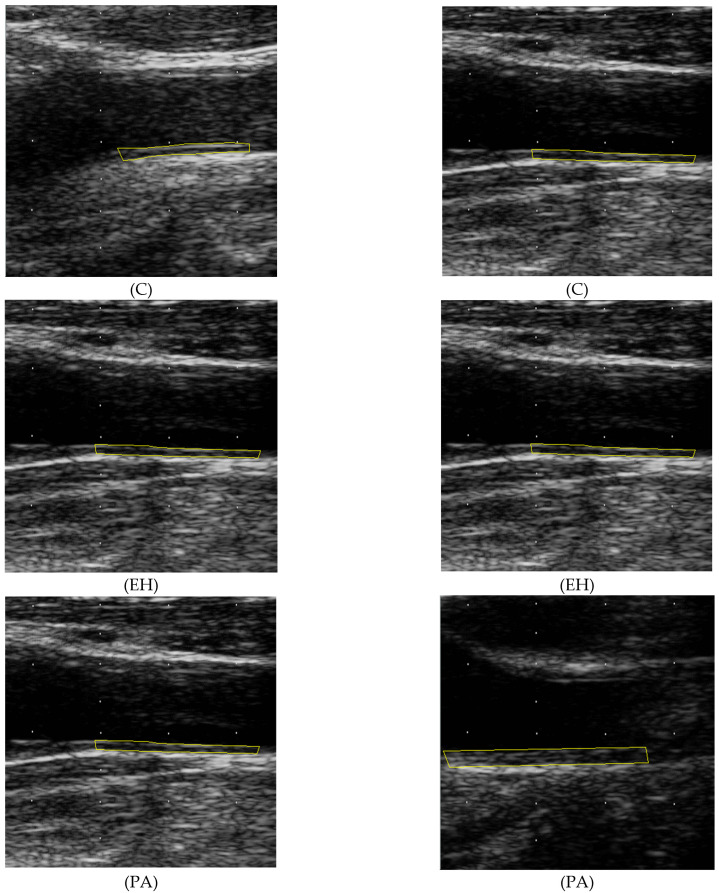
Example longitudinal carotid artery ultrasound images: healthy controls (C), patients with essential hypertension (EH), and primary aldosteronism (PA). The intima-media region of interest (ROI) on the left and right side is marked in yellow.

**Figure 4 diagnostics-12-03206-f004:**
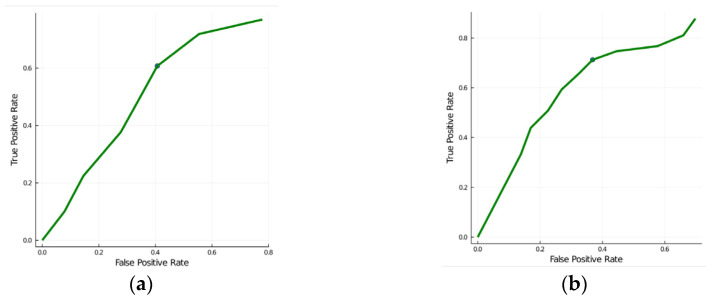
ROC curves for using texture features to distinguish between (**a**) PA versus EH and (**b**) PA + EH versus controls. C, controls; EH, essential hypertension; PA, primary aldosteronism.

**Table 1 diagnostics-12-03206-t001:** A list of 152 texture features and their symbols.

Feature	Description
*μ*	mean intensity (*n* = 1)
*σ*	standard deviation of the intensity (*n* = 1)
	Haralick texture descriptors (*n* = 5 × 4 × 7 = 140)
	*d* displacement in pixels (1, 2, 3, 4, and 5)
*H_d_* _,*θ*,*a*_	*θ* displacement direction (0°, 45°, 90°, and 135°)
	*a* texture descriptor (1: inverse difference moment, 2: correlation, 3: contrast, 4: maximum, 5: energy, 6: dissimilarity, and 7: entropy)
	Wavelet texture descriptors (*n* = (3 × 3) + 1 = 10)
	*i* decomposition level (1, 2, and 3)
*W_i_* _,*j*_	*j* filter orientation (1: G_x_G_x_, 2: H_x_G_y_, and 3: G_x_H_y_)
*W* _0_	low-pass subband wavelet energy

All intensities are evaluated inside the region of interest (ROI). H and G are the low-pass and high-pass Haar filters, respectively.

**Table 2 diagnostics-12-03206-t002:** Clinical characteristics and humoral data of the study groups.

	Primary Aldosteronism (*n* = 33)	Essential Hypertension (*n* = 52)	Hypertensive Patients (*n* = 85)	Normotensive Controls (*n* = 33)
Age (years)	57 ± 8	55 ± 7	56 ± 7	54 ± 8
Gender (F/M (%F))	13/20 (39)	20/32 (38)	33/52 (39)	16/17 (48)
Body mass index (kg·m^−2^)	29.0 ± 3.9 ^#^	28.4 ± 4.6 ^#^	**28.7±4.3 ^#^**	**26.6 ± 4.2**
Office systolic blood pressure (mm Hg)	164 ± 23 ^##^	170 ± 30 ^##^	**168±28 ^##^**	**125 ±15**
Office diastolic blood pressure (mm Hg)	97 ± 14 ^##^	99 ± 17 ^##^	**98±16 ^##^**	**78 ± 9**
Current cigarette smoking (*n* (%))	11 (33)	20 (38)	31 (36)	8 (24)
Lipid lowering medication (*n* (%))	10 (30)	21 (40)	31 (36)	9 (27)
Diabetes mellitus (*n* (%))	8 (24)	11 (21)	19 (22)	-
Plasma cholesterol (mmol/L)	4.98 ± 1.05 ^#^	5.28 ± 1.00 ^#^	**5.16 ± 1.02 ^#^**	**5.59 ± 0.97**
LDL cholesterol (mmol/L)	2.99 ± 0.87	3.06 ± 0.77	3.03 ± 0.81	3.19 ± 0.84
HDL cholesterol (mmol/L)	1.30 ± 0.34 ^#^	1.38 ± 0.33 ^#^	**1.35 ± 0.34 ^#^**	**1.61 ± 0.40**
Triglycerides (mmol/L)	1.54 ± 0.54	1.85 ± 0.97	1.73 ± 1.02	1.73 ± 1.02
Fasting plasma glucose (mmol/L)	4.9 ± 0.5	5.2 ± 0.9	5.1 ± 0.8	5.0 ± 0.6
Plasma potassium (mmol/L)	**3.6 ± 0.5 **^,##^**	**4.0 ± 0.3 ^##^**	**3.8 ± 0.4 ^##^**	**4.4 ± 0.4**
Plasma sodium (mmol/L)	143 ± 3	142 ± 3	143 ± 3	141 ± 2
Plasma aldosterone-upright (ng/dL)	**38.9 (27.4–64.9) ****	**15.7 (9.2–23.1)**		NA
Plasma renin activity-upright (ng/mL per h)	**0.36 (0.25–0.56) ****	**0.66 (0.37–1.68)**		NA
Aldosterone to plasma renin activity ratio-upright (ng/dL)/(ng/mL per h)	**103 (69–157) ****	**17 (8–41)**		NA
Urine potassium/day (mmol/24 h)	**62 (47–100) ***	**46 (31–58)**		NA
CCA IMT mean-max (mm)	**0.987 ± 0.152 *^,##^**	**0.892 ± 0.155 ^#^**	**0.93 ± 0.16 ^#^**	**0.812 ± 0.126**
CB IMT mean-max (mm)	1.157 ± 0.243 ^#^	1.131 ± 0.275 ^#^	**1.14 ± 0.26 ^#^**	**0.994 ± 0.203**
Combined IMT mean-max (mm)	1.066 ± 0.162 ^##^	0.974 ± 0.196 ^#^	**1.01 ± 0.19 ^#^**	**0.884 ± 0.141**

Variables are shown as means ± S.D., medians (interquartile range), or absolute numbers and percentages. CB, carotid bifurcation; CCA, common carotid artery; IMT, intima-media thickness; HDL, high-density lipoprotein; LDL, low density lipoprotein; NA, denotes unavailable measurements due to aldosterone and renin levels not being evaluated in healthy controls; NS, not significant; * *p* < 0.05, ** *p* < 0.001 primary aldosteronism vs. essential hypertension; # *p* < 0.05, ## *p* < 0.001 primary aldosteronism, essential hypertension, hypertensive patients vs. controls.

**Table 3 diagnostics-12-03206-t003:** Use of antihypertensive drugs and blood pressure levels in both hypertensive groups.

	Primary Aldosteronism (*n* = 33)	Essential Hypertension (*n* = 52)
Estimated duration of hypertension (years)	11.5 ± 7.8	14.5 ± 11.2
Chronic antihypertensive therapy		
Diuretics (*n* (%))	20 (61)	31 (60)
β-Blockers (*n* (%))	20 (61)	39 (75)
Calcium channel blockers (*n* (%))	28 (85)	41 (79)
Angiotensin-converting enzyme inhibitors (*n* (%))	19 (58)	33 (63)
Angiotensin receptor blockers (*n* (%))	15 (45)	18 (35)
α-Blockers (*n* (%))	8 (24)	17 (33)
Central agonists (*n* (%))	18 (55)	23 (44)
Aldosterone antagonists	-	-
Number of antihypertensive drugs	4.0 ± 1.5	4.0 ± 1.8
Blood pressure after discontinuation of therapy		
Office systolic blood pressure (mm Hg)	164 ± 23	170 ± 30
Office diastolic blood pressure (mm Hg)	97 ± 14	99 ± 17
Mean 24 h systolic blood pressure (mm Hg)	150 ± 14	148 ± 15
Mean 24 h diastolic blood pressure (mm Hg)	90 ± 8	89 ± 10

**Table 4 diagnostics-12-03206-t004:** A subset of clinical and IMT parameters significantly different between the PA and EH groups or between the PA + EH and control groups.

	PA Versus EH			PA + EH Versus Controls		
Parameter	*t*-Test	KS Test	MW Test	*t*-Test	KS Test	MW Test
Plasma aldosterone	<0.001	<0.001	<0.001	NA	NA	NA
ARR	<0.001	<0.001	<0.001	NA	NA	NA
Plasma renin activity	0.004	0.002	<0.001	NA	NA	NA
CCA–IMT	0.007	0.012	0.013	<0.001	<0.001	<0.001
Combined mean IMT	0.022		0.025	<0.001	0.008	0.001
Plasma cholesterol				0.041		
HDL cholesterol				0.002	0.004	<0.001
Systolic blood pressure				<0.001	<0.001	<0.001
Body mass index				0.024		0.019
Diastolic blood pressure				<0.001	<0.001	<0.001
CB–IMT				0.004		0.016

Analysis according to Welch’s unequal variance *t*-test, sorted by the *p* value of this test. We show the *p* values for the *t*-test, Kolmogorov–Smirnov (KS), and Mann–Whitney (MW) *U* tests comparing means for PA versus EH and PA + EH versus control groups. Empty fields mean that the difference is not significant, while NA denotes unavailable measurements due to aldosterone and renin levels not being evaluated in healthy controls. ARR, aldosterone to renin ratio; CB, carotid bifurcation; CCA, common carotid artery; EH, essential hypertension; HDL, high-density lipoprotein; IMT, intima-media thickness; PA, primary aldosteronism.

**Table 5 diagnostics-12-03206-t005:** Differences between the PA + EH and control groups in the first 10 texture descriptors.

	PA Versus EH			PA + EH Versus Controls		
Parameter	*t*-Test	KS Test	MW Test	*t*-Test	KS Test	MW Test
*W* _2,2_				<0.001	<0.001	<0.001
*W* _3,3_				<0.001	0.034	0.003
*W* _3,2_				<0.001	<0.001	<0.001
*W* _3,1_				<0.001	0.001	<0.001
*H* _1,0,2_				0.001	<0.001	<0.001
*H* _1,135,2_				0.001	<0.001	<0.001
*W* _2,3_				<0.001	0.017	0.005
*H* _2,135,2_				<0.001	<0.001	<0.001
*H* _2,0,2_				<0.001	<0.001	<0.001
*W* _0_				<0.001	<0.001	<0.001

We show the *p* values for the *t*-test, Kolmogorov–Smirnov (KS), and Mann–Whitney (MW) *U* tests comparing PA versus EH and for PA + EH versus control groups. No texture feature alone was significant in distinguishing PA versus EH, as marked by empty fields. EH, essential hypertension; PA, primary aldosteronism. Empty fields mean that the difference is not significant.

**Table 6 diagnostics-12-03206-t006:** Per-patient classification accuracy evaluated by leave-one-out for different feature subsets and the two classification tasks.

	Accuracy	
Parameters	PA Versus EH	PA + EH Versus Controls
Clinical characteristics	0.57	0.89
Clinical characteristics + IMT parameters	0.63	0.92
Texture features	0.73	0.66
“Aldo” parameters	0.92	*NA*

EH, essential hypertension; IMT, intima-media thickness; PA, primary aldosteronism.

## Data Availability

The data is not publicly available due to patient privacy regulations.

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
