# Peer review of "The Effect of Primary Aldosteronism on Carotid Artery Texture in Ultrasound Images"

_diagnostics, 2022, doi:10.3390/diagnostics12123206_

Round 1
Reviewer 1 Report (Previous Reviewer 2)
The main thesis of this paper is that carotid intima-media thickening in patients with primary aldosteronism (PA) is significantly stronger than in patients with essential hypertension (EH) and controls. Although the results are well accepted and reflect organ damage in PA, it seems unreasonable to incorporate this carotid echo finding into diagnosing PA.
#1 It seems possible that the carotid artery echocardiographic thickening in patients with PA could be greater than in patients with EA, but the difference is only 0.1 mm. It seems difficult to distinguish PA from essential hypertension of this difference. If it is possible to distinguish between the two groups, the authors should show the distribution of the two groups in a figure so that we can understand the difference. Also, the authors should determine the cutoff value by ROC analysis, etc., and show how well you can distinguish between the two groups.
#2 Within PA, there are two broad categories: unilateral aldosterone-producing adenoma (APA) and bilateral idiopathic hyperaldosteronism (IHA), which are thought to differ in aldosterone levels and prognosis. In the present study, APA and IHA are thought to have different blood pressure and effects on carotid artery echocardiography.
#3 In general, PA patients seem to be younger than normal hypertensive patients, but their age is exactly the same in this study. How are essential hypertensive patients selected in this study? If there are selection criteria, please indicate them.
#4 Carotid echocardiography is an excellent method to evaluate carotid artery stiffness and can be performed easily and repeatedly. However, no matter how accurately one tries to measure, inter-observer variation is still present and needs to be accounted for.
Author Response
Comments and Suggestions for Authors
The main thesis of this paper is that carotid intima-media thickening in patients with primary aldosteronism (PA) is significantly stronger than in patients with essential hypertension (EH) and controls. Although the results are well accepted and reflect organ damage in PA, it seems unreasonable to incorporate this carotid echo finding into diagnosing PA.
I am afraid you did not quite understand the aim of this manuscript. You confused the assessment of the texture features of the common carotid arterial wall with the assessment of the IMT of the common carotid artery.
#1 It seems possible that the carotid artery echocardiographic thickening in patients with PA could be greater than in patients with EA, but the difference is only 0.1 mm. It seems difficult to distinguish PA from essential hypertension of this difference. If it is possible to distinguish between the two groups, the authors should show the distribution of the two groups in a figure so that we can understand the difference. Also, the authors should determine the cutoff value by ROC analysis, etc., and show how well you can distinguish between the two groups.
Difference of 0,1 mm is of course small, and it surely cannot be routinely used to differentiate between patients with PA and essential hypertension. Below the text, you can find boxplots which list the values of IMT in both hypertensive groups. Be that as it may, increased IMT during routine examination of the patients with resistant hypertension should lead to suspicion of PA as the most common form of secondary hypertension [2]. Given the low reliability of IMT measurement as the distinguisher between PA and EH patients we focused on more detailed analysis of the ultrasound image of the common carotid artery in order to find further descriptors, which would allow us to identify the PA patients with more certainty.
#2 Within PA, there are two broad categories: unilateral aldosterone-producing adenoma (APA) and bilateral idiopathic hyperaldosteronism (IHA), which are thought to differ in aldosterone levels and prognosis. In the present study, APA and IHA are thought to have different blood pressure and effects on carotid artery echocardiography.
Our cohort included 33 patients with PA. Sixteen patients were classified as aldosterone-producing adenoma (APA), 4 as idiopathic hyperaldosteronism (IHA) and 13 patients remained unclassified. It is statistically impossible to evaluate a group of 4. When joined together the IHA + unclassified group comprised 17 patients, this group did not differ in age, systolic BP, percentage of women or the IMT value measured on the common carotid artery when compared to the group of clearly diagnosed APA (0.97 ± 0.12 mm v.s. 1.00 ± 0.18 mm; p=0.69).
#3 In general, PA patients seem to be younger than normal hypertensive patients, but their age is exactly the same in this study. How are essential hypertensive patients selected in this study? If there are selection criteria, please indicate them.
Our study included patients with resistant hypertension who had been referred to our center for resistant hypertension with the diagnosis of suspected secondary hypertension. The diagnosis of PA was confirmed in one fifth of the patients. In four fifths of the remaining patients, some patients were excluded due to other forms of secondary hypertension, white coat hypertension and noncompliance with antihypertensive therapy. The remaining cohort of patients with “simple” EH did not differ in age, percentage of women and systolic BP [3]. The relatively lower age in the group with essential hypertension when compared to normal hypertensive population could be due to the selection for referral to our center. Older hypertensive patients may not be referred to our center as often for example due to a lower probability of secondary hypertension diagnosis. This paragraph has been added to the revised version of the manuscript.
#4 Carotid echocardiography is an excellent method to evaluate carotid artery stiffness and can be performed easily and repeatedly. However, no matter how accurately one tries to measure, inter-observer variation is still present and needs to be accounted for.
The inter- and intraindividual variability has been evaluated in our studies regarding the IMT topic repeatedly. Namely in the study ultrasound examinations and the off-line IMT measurements were performed by a single sonographer and reader in one person (R.H.) blinded to participants´ diagnoses. Ultrasound examinations were performed in duplicate in 20 subjects within 3 weeks. IMT reproducibility quantified by interclass correlation coefficient was 0.86 for the distal segment of common carotid artery (CCA) and 0.79 for the segment of carotid bifurcation (CB) [1]. This paragraph has been added to the revised version of the manuscript.
References
- Holaj, R.; Zelinka, T.; Wichterle, D.; Petrák, O.; Štrauch, B.; Widimský, J., Jr. Increased intima-media thickness of the common carotid artery in primary aldosteronism in comparison with essential hypertension. Journal of hypertension 2007, 25, 1451-1457, doi:10.1097/HJH.0b013e3281268532.
- Štrauch, B.; Zelinka, T.; Hampf, M.; Bernhardt, R.; Widimský, J., Jr. Prevalence of primary hyperaldosteronism in moderate to severe hypertension in the Central Europe region. Journal of human hypertension 2003, 17, 349-352, doi:10.1038/sj.jhh.1001554.
- Rosa, J.; Zelinka, T.; Petrák, O.; Štrauch, B.; Šomloová, Z.; Indra, T.; Holaj, R.; Čurila, K.; Toušek, P.; Senítko, M.; et al. Importance of thorough investigation of resistant hypertension before renal denervation: should compliance to treatment be evaluated systematically? Journal of human hypertension 2014, 28, 684-688, doi:10.1038/jhh.2014.3.

Reviewer 2 Report (Previous Reviewer 1)
The problem is interesting to solve, however, the methodology is quiet confusing especially after adding the block diagram as it was not mentioned in the previous review. The answers of most of the comments are not convincing. The manuscript has some fundamental weakness such as the methodology, assessment criterion of the approach, and comparison etc. The methodology (Figure 3) includes four classifiers, and it is unclear that how the result is achieved from these four individual models. ROI selection and ROI extraction are not clear and how authors first select ROI and apply some pre-processing steps and then extract ROI which is quite confusing. The IMT is measured after ROI selection, and authors applied IMT measurements before it, as mentioned in the block diagram. Biomedical and clinical attributes have independent classifiers without justification of final decision. At line 214, authors mentioned the use of majority vote, and it is unclear that among four classifiers, tie breaking is not mentioned.
Author Response
Comments and Suggestions for Authors
The problem is interesting to solve, however, the methodology is quiet confusing especially after adding the block diagram as it was not mentioned in the previous review. The answers of most of the comments are not convincing. The manuscript has some fundamental weakness such as the methodology, assessment criterion of the approach, and comparison etc.
The methodology (Figure 3) includes four classifiers, and it is unclear that how the result is achieved from these four individual models.
This is probably a misunderstanding. We are not using all four classifiers. Our method only uses one classifier based on Haralick and texture features. The other three classifiers are only used in the experiments to serve as a baseline for our method. Graphically, the new parts that our method consists of are framed in bold, while the steps done previously or done only for the experimental comparison have a think frame. This paragraph has been added to the revised version of the manuscript.
ROI selection and ROI extraction are not clear and how authors first select ROI and apply some pre-processing steps and then extract ROI which is quite confusing.
We are sorry for the confusion. This is described in Section 2.4 and 2.5. The "ROI selection" is performed manually. Image pre-processing and ROI extraction could probably be grouped together, they consist of intensity normalization, scale normalization, interpolation and cropping.
The IMT is measured after ROI selection, and authors applied IMT measurements before it, as mentioned in the block diagram.
This is probably the same misunderstanding as mentioned above. IMT is measured by an expert, and it is not part of our method. We only use it as a baseline for comparison of the performance of our image-based method.
Biomedical and clinical attributes have independent classifiers without justification of final decision. At line 214, authors mentioned the use of majority vote, and it is unclear that among four classifiers, tie breaking is not mentioned.
This is related to the same misunderstanding. Our method only uses one classifier. As described in Section 2.6, the majority voting is done over images (since we have multiple images per patient) not classifiers.

Round 2
Reviewer 1 Report (Previous Reviewer 2)
I have no further comments regarding the revised manuscript.
Author Response
We have tried to correct all the grammatical and typo mistakes.

Reviewer 2 Report (Previous Reviewer 1)
Authors considered some of the points and incorporated in the revised version. However, the fundamental questions are still pending. For instance, Figure 3, must be clear that which method/classifier decision is being used by the author. Authors described texture and Wavelet features for classification, but unable to explain how many features of are extracted. ROI selection is another challenge that how it is segregated from the background. Comparison is not available. IMT plot is necessary to know the difference between a healthy and a diseased subject. Technical, grammatical and typo mistakes are very common for instance, at line 209, authors mentioned that “the features are used to rain an XGBoost classifier for both tasks”. Comparison of the authors and other approaches is not available. In Figure 3, authors need to mention the classifier name instead of ‘classifier’ in a box. Moreover, ROC curve must be plotted for the assessment of the model. Table 1 is not useful especially in the absence of mathematical equations. Either authors used mathematical equations and describes the used symbols, or they refer the texture features reference by mentioning the number of features and their rational of its use. Overall, the work is still very weak, and it must be improved significantly as per instructions provided above.
Author Response
Comments and Suggestions for Authors
Authors considered some of the points and incorporated in the revised version. However, the fundamental questions are still pending.
For instance, Figure 3, must be clear that which method/classifier decision is being used by the author.
- As classifier, the XGBoost was used. It is noted in the new version of the manuscript.
Authors described texture and Wavelet features for classification, but unable to explain how many features of are extracted.
- One hundred and fifty-two texture features were extracted. Two texture features are the intensity mean and standard deviation, one hundred and forty are Haralick texture descriptors and ten are wavelet texture descriptors. This information is noted in Section 2.5 in the new version of the manuscript.
ROI selection is another challenge that how it is segregated from the background. Comparison is not available.
- Automatic ROI selection is indeed a challenging process, but it is not the focus of this manuscript. As described in Section 2.5, the ROI was drawn by the technician using a custom software. The ROI copied the boundaries selected by the program Image Pro-Plus and was afterwards checked by a physician experienced in assessment of ultrasound findings (B.M.) to minimize the observer variability. In each image of the distal part of common carotid artery, the visualized blood-intima and media-adventitia boundaries of the far wall were marked within the minimal 10 mm length segment. The last sentence is included in the new version of the manuscript.
IMT plot is necessary to know the difference between a healthy and a diseased subject.
- We are not sure what is meant by the "IMT plot". This manuscript is not about using IMT, this was described in previous work [20]. This manuscript focuses on the use of texture features for the classification. However, IMT is evaluated (as mentioned in Section 2.4) and it is used as a reference in the experimental comparison in Section 2.5.
Technical, grammatical and typo mistakes are very common for instance, at line 209, authors mentioned that “the features are used to rain an XGBoost classifier for both tasks”.
- We have tried to correct all the grammatical and typo mistakes. Specifically, in that sentence, the consonant "t" fell out in the word "train".
Comparison of the authors and other approaches is not available.
- To our knowledge, there is no other work using carotid artery wall texture to predict primary aldosteronism, so a direct comparison is not possible. However, we do compare our prediction based on texture with predictions based on other descriptors in Section 2.5.
In Figure 3, authors need to mention the classifier name instead of ‘classifier’ in a box.
- As classifier, the XGBoost was used. It is noted in the new version of the manuscript.
Moreover, ROC curve must be plotted for the assessment of the model.
- ROC curves were added into the new version of the manuscript.
Table 1 is not useful especially in the absence of mathematical equations. Either authors used mathematical equations and describes the used symbols, or they refer the texture features reference by mentioning the number of features and their rational of its use.
- Mathematical equations were added into the new version of the manuscript.
Overall, the work is still very weak, and it must be improved significantly as per instructions provided above.
- Thank you for your patience and your valuable comment. We apologize to you for the misunderstanding in previous round.

Round 3
Reviewer 2 Report (Previous Reviewer 1)
Review:
The manuscript is improved compared to its previous versions and authors are working hard to improve their manuscript. However, still there are several important points left that must be improved before the manuscript proceed further. These points are given below:
1. In abstract of the manuscript, authors mentioned that they are using support vector machine (SVM) for classification, whereas it never used in the whole manuscript. Authors must carefully check and review to end such inconsistencies.
2. The claim of 73% accuracy on PA and EH classes is not justified from Figure 5 (a) (ROC curve). It might be possible that there is any discrepancy in drawing ROC curve, authors need to check it very carefully.
3. From Figure 4(a), the inconsistency of Figure 4(a) and accuracy value of 73% show that XGBoost was not able to learn the pattern successfully. Such type of inconsistency possible when model is unable to identify most of the data sample of a class. Moreover, for better understanding, authors need to compute other parameters such as Sensitivity, Specificity, F-Score, Negative Predictive Value (NPV), and Positive Predictive Value (PPV), and Matthew correlation coefficient (MMC). If these parameters are undervalued, then authors must use another classifier instead of XGBoost.
4. In abstract, authors mentioned that the accuracy value is 57% better than IMT alone. This point is not justified in the results, and discussion section. Authors need to clarify it because the claim is very much high, and it needs proper discussion.
5. Again on Figure 2, my question of previous version is not properly answered. Authors mentioned some blocks green and where the confusion of ROI selection and ROI extraction persists. Authors remove the ROI extraction box because after preprocessing, features are extracted. Moreover, the IMT measurement input arrow must be from ROI selection. Because once ROI is identified by the expert, the IMT measurement process also performed on it.
6. Authors required to assign numbers to mathematical equations and refer in manuscript where needed.
7. In Table 1, the number of features are exceeding 152. Authors are required to cross check and write correct number.
8. At line 257 of page 7, authors are required to mention task 1 and 2 against each number.
9. In last paragraph of Section 3, authors are required to mention the distribution of images i.e. how many images per subject were collected and used for model training.
10. In conclusion, line 422, authors mentioned that although the prediction accuracy is low but useful for clinical practices. To justify this point, author need to discuss scenarios in discussion section so that readers can understand the significance of the proposed approach.
11. Still there are some typo and grammatical mistakes which need careful proofreading.
This manuscript is a resubmission of an earlier submission. The following is a list of the peer review reports and author responses from that submission.
Round 1
Reviewer 1 Report
Review:
Authors examine the effects of hyperaldosteronism carotid artery effects by employing texture features and SVM classification. Although the topic is interesting, however, the manuscript has structural weaknesses. The carotid artery examination and plaque detection is well established area of research in the literature. The novelty of the manuscript is not clear, in fact which is very low. Reviewer feel that the manuscript needs significant improvement before further processing. Specific comments are as follows.
1. Selection of ROI (region of interest) is a fundamental question that how authors selected it? Currently, it seems it is a selected manually that may raise the questions of how it will assist radiologists where the analysis is fully user dependent?
2. Authors mentioned that early diagnosis of PA may avoid several complications which is true. However, authors are unable to explain it in their results and there is no evidence of early detection measurements used in this research to support their claim.
3. The carotid plaque requires IMT values which required to be segregated from the background and other tissues. However, authors are unable to explain this concept and directly jumped to simple classification by selecting ROI manually and fed to SVM.
4. Ultrasound images have some built-in noise types which may have negatively impact on classification results. Authors need to explain that how they avoid/overcome the noise issue in their research?
5. Block diagram/flow chart of the methodology is missing which may affect on the reproduction of the results by the other researchers in case they interested to extend it.
6. Instead of image features (as authors mentioned the texture features), authors used the clinical features which are mentioned in Sec. 3.4, which I think are not supporting the research hypothesis. Because authors without any rational selects these 11 clinical features by rejecting the image features.
7. Continuing Q. 6, if the clinical features are only used then there is absolutely no need of carotid artery ultrasound. Authors need to justify the ultrasound images required for this diagnosis.
8. There is no rational given for leave one out method for model training. Moreover, authors are unable to explain why just SVM is used and there is no comparison of results which shows the deficiency of research.
9. See some of the research papers used for carotid artery plaque detection and classification. Robust Hidden Markov Model based intelligent blood vessel detection of fundus images, Robust spatial fuzzy GMM based MRI segmentation and carotid artery plaque detection in ultrasound images.
10. Discussion of the manuscript is very weak and there are some grammatical and typo mistakes which need a thorough revision.
11. Some images of patient are included (Figure 3), I am not sure authors have permission to include that image.
Reviewer 2 Report
The main thesis of this paper is that carotid intima-media thickening in patients with primary aldosteronism (PA) is significantly stronger than in patients with essential hypertension (EH) and controls. While the results are acceptable, the interpretation of the results seems difficult to accept.
#1 It seems possible that the carotid artery echocardiographic thickening in patients with PA could be greater than that in patients with EA, but the difference is only 0.1 mm. It seems difficult to distinguish primary aldosteronism from essential hypertension by this difference. If it is possible to distinguish between the two groups, the authors should show the distribution of the two groups in a figure so that we can understand the difference. Also, the authors should determine the cutoff value by ROC analysis, etc., and show how well you can distinguish between the two groups.
#2 Carotid echocardiography is an excellent method to evaluate carotid artery stiffness and can be performed easily and repeatedly. However, no matter how accurately one tries to measure, inter-observer variation is still present and needs to be accounted for.
#3 Since the significant differences in Table 2 are not easy to understand, and I think it would be easier to understand if the significant differences were shown separately for PA vs. EH and PA+EH vs. Control.
#4 Any comments on the differences in blood glucose?